# Whole Blood Fatty Acid Profiles of Cold-Stunned Juvenile Green, Kemp's Ridley, and Loggerhead Sea Turtles

Ashlyn C. Heniff [1], Larry J. Minter [1,2,3], Craig A. Harms [1,2,4], Doug Bibus [5], Elizabeth A. Koutsos [6] and Kimberly D. Ange-van Heugten [2,7,*]

1    Department of Clinical Sciences, College of Veterinary Medicine, North Carolina State University, Raleigh, NC 27606, USA
2    Environmental Medicine Consortium, North Carolina State University, Raleigh, NC 27606, USA
3    North Carolina Zoo, Asheboro, NC 27205, USA
4    Center for Marine Sciences and Technology, North Carolina State University, Morehead City, NC 28557, USA
5    Lipid Technologies, LLC, P.O. Box 216, Austin, MN 55912, USA
6    EnviroFlight, LLC, 1118 Progress Way, Maysville, KY 41056, USA
7    Department of Animal Science, College of Agriculture and Life Sciences, North Carolina State University, Raleigh, NC 27695, USA
*    Correspondence: kim_ange@ncsu.edu

**Abstract:** When subjected to cold environmental temperatures, cheloniid sea turtles can experience debilitating lethargy, anorexia, and potential mortality in a phenomenon known as cold-stunning. Every year, hundreds to thousands of cold-stunned sea turtles are transported to rehabilitation centers for medical and nutritional care. The objective of this study was to investigate one aspect of nutritional status in cold-stunned sea turtles: fatty acid profiles. Blood was collected from eleven green (*Chelonia mydas*), twelve Kemp's ridley (*Lepidochelys kempii*), and three loggerhead (*Caretta caretta*) juvenile sea turtles found cold-stunned along the coast of North Carolina, USA. Whole blood (~160 µL) was dried onto specialized paper spot cards, frozen, and subsequently analyzed via gas chromatography to quantify fatty acid percentages. Significant differences among species were identified for 19 out of 36 individual fatty acids analyzed and six out of seven fatty acid groups evaluated ($P < 0.5$). The whole blood fatty acid profiles of cold-stunned green and Kemp's ridley sea turtles were similar to prior published profiles of healthy conspecifics. Marginal numerical differences noted upon visual comparison included that cold-stunned sea turtles had lower proportions of total polyunsaturated fatty acids (PUFA) and monoenes and higher proportions of total saturated fatty acids relative to healthy conspecifics. These differences may reflect acute impacts of cold-stunning on circulating plasma fatty acids or may be the result of natural seasonal variations. These data provide practical information to aid in the diet design of sea turtles in rehabilitation settings.

**Keywords:** fatty acids; cold-stunning; nutrition; sea turtles

## 1. Introduction

Loggerhead (*Caretta caretta*), green (*Chelonia mydas*), and Kemp's ridley (*Lepidochelys kempii*) sea turtles are listed as vulnerable, endangered, and critically endangered species, respectively, by the International Union for the Conservation of Nature [1–3]. As poikilotherms, these hard-shelled (cheloniid) sea turtles rely heavily on their external environment for internal temperature regulation. Prolonged exposure to relatively cold temperatures (12 °C or lower) can induce cold-stun syndrome: a lethargic or moribund state characterized by anorexia, positive buoyancy, and bradycardia [4,5]. Although cold-stunning has been reported since the 1800s and is considered a natural phenomenon, warming ocean temperatures may be exacerbating morbidity by modifying the northerly distribution of juvenile sea turtles [6]. Every winter, hundreds to thousands of cold-stunned sea turtles

found along the western Atlantic coast are transported to rehabilitation centers [4,6]. When provided with appropriate medical and nutritional care, many of these animals recover and are wild-released within days to months, depending upon weather conditions and case severity [4,7].

Respiratory and/or metabolic acidosis, dehydration, and renal dysfunction are commonly reported in cold-stunned sea turtles; thus, blood gas, electrolyte, and biochemical analyses are critical to formulating appropriate therapeutic regimens [8–12]. To further examine the pathogenesis of cold-stunning and its potential nutritional ramifications, a recent study used nuclear magnetic resonance (NMR) spectroscopy to establish baseline metabolomic profiles of cold-stunned green, Kemp's ridley, and loggerhead sea turtles [13]. Investigators identified several differences in polar (aqueous) metabolite concentrations between cold-stunned sea turtles and presumably healthy conspecifics, but the technology employed did not permit the assessment of non-polar (lipophilic) molecules, such as fatty acids [13].

The regulation of fatty acid metabolism is essential for all animals, as these molecules play important roles in growth and development, immune response, organ function, reproductive success, and cellular integrity and signaling [14–17]. Fatty acid concentrations and availability in the body are directly influenced by dietary intake and can be altered in disease states [16–18]. Consequently, fatty acid profiles are a focal point for the development of evidence-based diets for animals in managed care settings. Dried blood spot cards and subsequent gas chromatography allow for the determination of fatty acid profiles from just two to three drops of whole blood [19–28]. These spot cards are easy to transport and do not require immediate freezing or refrigeration, making them convenient for use in field settings. Recent studies have successfully established fatty acid profiles using these spot cards for presumably healthy juvenile green and Kemp's ridley sea turtles in wild and managed care settings [22,23], but to the authors' knowledge, no prior studies have evaluated fatty acids in cold-stunned sea turtles.

It is important to note that whole blood fatty acid profiles are derived from a blend of fatty acids contained in plasma and circulating cells [29]. In humans, plasma fatty acid profiles reflect short-term fatty acid status, while erythrocyte fatty acid profiles are better suited for assessing long-term fatty acid metabolism, as human erythrocytes have rather long lifespans (~120 days) and their lipid turnover is much slower than that of plasma [29–33]. Thus, whole blood fatty acid profiles may provide a balanced picture of long-term and short-term fatty acid status [29]. Although several studies have used dried blood spot cards to assess fatty acid status in reptiles [20–23], the extent to which reptilian whole blood fatty acid profiles can be expected to reflect acute alterations in fatty acid status has not been extensively investigated.

The evaluation of the whole blood fatty acid profiles of cold-stunned sea turtles may allow for the improved medical and nutritional management of affected sea turtles in rehabilitation settings, ultimately translating to a higher likelihood of release and/or more rapid recoveries. Furthermore, this information may advance understanding of the potential effects of cold-stunning on sea turtle metabolomics and associated physiological consequences. The objective of this study was to examine whole blood fatty acid profiles of cold-stunned green, Kemp's ridley, and loggerhead sea turtles using dried blood spot cards and compare these among species and to previously established profiles in healthy conspecifics.

## 2. Materials and Methods

Eleven green, twelve Kemp's ridley, and three loggerhead sea turtles of unknown sex found cold-stunned along the coast of North Carolina, USA were included. All animals were classified as juvenile based on carapace length and the absence of secondary sex characteristics [34,35]. Green and Kemp's ridley sea turtles were rescued 29–30 January of 2021 and 11–28 January of 2022, respectively; loggerhead sea turtles were rescued 30 January 2021, 4 February 2021, and 28 January 2022. Upon rescue, the turtles were transferred to a nearby triage location where weight, length, and cloacal temperature were

recorded, and venipuncture was performed via the external jugular vein (dorsal cervical sinus) using an appropriately sized heparinized needle and syringe. Two ~80 μL whole blood samples were immediately applied to a Perkin-Elmer Spot Saver Card (Perkin-Elmer Health Sciences, Inc., Greenville, SC, USA) and allowed to air dry (15 minutes). Within 4 hours, the spot cards were stored at −80 °C.

The spot cards were sent out within 2 months of collection for fatty acid analysis (Lipid Technologies, Austin, MN, USA; https://lipidlab.com/services/) using a previously established methodology [22,23,25–28,36] in which blood spots were transmethylated with acidified methanol, and fatty acid methyl esters were then analyzed by gas chromatography (NuChek Prep FAME std 490, Nu-Chek Prep, Elysian, MN, USA; nu-chekprep.com). In total, 36 individual fatty acids were analyzed and 7 fatty acid groups were evaluated. As sample volume was not quantified, fatty acid percentages, rather than quantities, are reported.

Following sampling, sea turtles were transferred to the Karen Beasley Sea Turtle Rescue and Rehabilitation Center (Surf City, North Carolina 28445, USA) for appropriate care. Animal capture, sample collection, and transfer were authorized by the North Carolina Wildlife Resources Permit 21ST42 and 22ST42 and North Carolina State University IACUC protocol 20-166-01.

## 3. Statistical Analysis

Data were analyzed for differences by species, with statistical significance set at $p < 0.05$, using Proc GLM procedures of SAS 9.4 (Cary, NC, USA). The least-square means and standard error of the means (SEM) were calculated and are included in the model statement. Visual comparisons were made with previously published values for healthy free-ranging juvenile green and Kemp's ridley sea turtles [19]. No whole blood fatty acid profiles of healthy juvenile loggerhead sea turtles were available for comparison.

## 4. Results

Median and range weights, straight carapace lengths (nuchal notch to pygal notch), and cloacal temperatures upon arrival for each species are displayed in Table 1. The whole blood fatty acid profiles of cold-stunned juvenile green, Kemp's ridley, and loggerhead sea turtles are presented in Table 2 alongside prior-published whole blood fatty acid profiles of apparently healthy juvenile green and Kemp's ridley sea turtles [22]. All seven fatty acid groupings were identified in the analysis for each species. Of the 36 individual fatty acids measured, 7 were below the limits of detection in all species.

**Table 1.** Size and cloacal temperatures upon arrival to the triage location for cold-stunned juvenile green (*Chelonia mydas*), Kemp's ridley (*Lepidochelys kempii*), and loggerhead (*Caretta caretta*) sea turtles sampled for whole blood fatty acid profiles.

| | *C. mydas* (n = 11) | | *L. kempii* (n = 12 *) | | *C. caretta* (n = 3) | |
|---|---|---|---|---|---|---|
| | **Median** | **Range** | **Median** | **Range** | **Median** | **Range** |
| **Weight (kg)** | 3.4 | 3.0–18.4 | 2.3 | 1.7–5.8 | 35.0 | 23.2–49.0 |
| **SCL-N # (cm)** | 30.8 | 29.2–50.7 | 26.3 | 23.2–36.3 | 64.0 | 56.9–74.0 |
| **Temperature (°C)** | **5.8** | **4.4–8.0** | **11.1** | **7.8–12.8** | **6.7** | **4.6–8.9** |

* Arrival temperature data unavailable for 2 Kemp's ridley sea turtles. # Straight carapace length (nuchal notch to pygal notch).

**Table 2.** Total whole blood individual fatty acid and fatty acid grouping profiles (%, mean ± SEM) for cold-stunned juvenile green (*Chelonia mydas*), Kemp's ridley (*Lepidochelys kempii*), and loggerhead (*Caretta caretta*) sea turtles. Results are quantified by area % and values are provided as a percent of total fatty acids present. Published profiles of healthy free-ranging juvenile green and Kemp's ridley sea turtles are presented for comparison [Koutsos et al., 2021].[22] ND indicates not detected; NR indicates not reported.

| | | Cold-Stunned Sea Turtle Fatty Acid Profiles | | | | Published Healthy Sea Turtle Fatty Acid Profiles | |
|---|---|---|---|---|---|---|---|
| *Individual Fatty Acids* * | | *Species* | | | *p*-Value [#] | *Species* | |
| **Fatty Acid** | **Common Name** | *C. mydas* (n = 11) | *L. kempii* (n = 12) | *C. caretta* (n = 3) | | *C. mydas* (n = 9) | *L. kempii* (n = 8) |
| 14:0 | Myristic acid | 5.2 ± 0.45 [a] | 6.9 ± 0.44 [b] | 4.9 ± 0.87 [a] | 0.0251 | 4.2 ± 0.50 | 5.0 ± 0.41 |
| 14:1 | Myristoleic acid | 0.04 ± 0.14 [a] | 0.49 ± 0.13 [b] | 1.72 ± 0.26 [c] | <0.0001 | NR | NR |
| 15:0 | Pentadecylic acid | 0.37 ± 0.10 [a] | 0.83 ± 0.09 [b] | 0.86 ± 0.19 [b] | 0.0061 | ND | ND |
| 16:0 | Palmitic acid | 20.0 ± 0.46 [a] | 19.8 ± 0.44 [a] | 15.3 ± 0.88 [b] | 0.0002 | 17.4 ± 0.59 | 16.9 ± 0.37 |
| 16:1n7 | Palmitoleic acid | 4.5 ± 0.77 [a] | 15.8 ± 0.74 [b] | 10.0 ± 1.48 [c] | <0.0001 | 5.4 ± 0.46 | 12.0 ± 0.56 |
| 17:0 | Margaric acid | ND | ND | ND | – | 0.4 ± 0.12 | 0.9 ± 0.06 |
| 17:1n7 | Heptadecenoic acid | ND | ND | ND | – | 0.8 ± 0.20 | 2.6 ± 0.25 |
| 18:0 | Stearic acid | 12.5 ± 0.58 [a] | 7.5 ± 0.56 [b] | 9.0 ± 1.12 [b] | <0.0001 | 10.9 ± 0.25 | 8.5 ± 0.37 |
| 18:1n7 | Vaccenic acid | 2.7 ± 0.24 [a] | 3.7 ± 0.23 [b] | 3.8 ± 0.45 [b] | 0.0135 | 2.2 ± 0.69 | 3.8 ± 0.54 |
| 18:1n9 | Oleic acid | 25.5 ± 0.80 [a] | 20.8 ± 0.76 [b] | 23.3 ± 1.53 [a,b] | 0.0012 | 28.3 ± 1.24 | 23.6 ± 0.73 |
| 18:2n6 | Linoleic acid | 4.6 ± 0.59 | 3.6 ± 0.56 | 3.7 ± 1.12 | 0.4621 | 4.9 ± 0.51 | 3.2 ± 0.14 |
| 18:3n6 | γ-Linoleic acid | 0.04 ± 0.024 [a] | 0.17 ± 0.023 [b] | 0.13 ± 0.046 [b] | 0.0034 | 0.1 ± 0.03 | 0.1 ± 0.01 |
| 18:3n3 | α-Linolenic acid | 1.0 ± 0.13 | 0.7 ± 0.12 | 0.6 ± 0.25 | 0.2183 | 1.5 ± 0.47 | 0.6 ± 0.06 |
| 18:4n3 | Stearidonic acid | 0.17 ± 0.037 | 0.17 ± 0.035 | 0.27 ± 0.071 | 0.3813 | NR | NR |
| 20:0 | Arachidic acid | 0.22 ± 0.023 [a] | 0.18 ± 0.022 [b] | 0.33 ± 0.044 [b] | 0.0297 | 0.1 ± 0.03 | 0.2 ± 0.03 |
| 20:1n7 | Paullinic acid | 0.76 ± 0.087 [a] | 0.30 ± 0.083 [b] | 0.76 ± 0.117 [a] | 0.0020 | 0.5 ± 0.11 | 0.6 ± 0.04 |
| 20:1n9 | Gondoic acid | ND | ND | ND | – | 0.06 ± 0.02 | 0.1 ± 0.02 |
| 20:2n6 | Eicosadienoic acid | 0.30 ± 0.046 | 0.19 ± 0.044 | 0.32 ± 0.087 | 0.2047 | 0.1 ± 0.04 | 0.4 ± 0.02 |
| 20:3n3 | Eicosatrienoic acid | 0.04 ± 0.013 | 0.01 ± 0.013 | 0.04 ± 0.026 | 0.2936 | 0.1 ± 0.02 | 0.1 ± 0.02 |
| 20:3n9 | Mead acid | 0.08 ± 0.020 [a] | 0.00 ± 0.019 [b] | 0.15 ± 0.038 [a] | 0.0018 | 0.5 ± 0.26 | 0.0 ± 0.02 |
| 20:3n6 | Dihomo-γ-Linoleic acid | 0.64 ± 0.059 [a] | 0.21 ± 0.056 [b] | 0.39 ± 0.123 [b] | <0.0001 | 0.6 ± 0.06 | 0.4 ± 0.02 |
| 20:4n6 | Arachidonic acid | 13.5 ± 1.04 [a] | 9.5 ± 0.99 [b] | 13.4 ± 1.98 [a,b] | 0.0232 | 12.8 ± 1.42 | 11.7 ± 0.50 |
| 20:4n3 | Eicosatetraenoic acid | 0.06 ± 0.040 | 0.17 ± 0.039 | 0.09 ± 0.078 | 0.1848 | 0.1 ± 0.03 | 0.1 ± 0.02 |
| 20:5n3 | Eicosapentaenoic acid | 1.9 ± 0.33 [a] | 4.1 ± 0.31 [b] | 2.7 ± 0.63 [a,b] | 0.0003 | 2.1 ± 0.14 | 4.0 ± 0.31 |
| 22:0 | Behenic acid | 0.15 ± 0.027 | 0.12 ± 0.026 | 0.25 ± 0.051 | 0.0880 | 0.1 ± 0.01 | 0.2 ± 0.02 |
| 22:1n9 | Erucic acid | 0.02 ± 0.042 [a] | 0.32 ± 0.040 [b] | 0.29 ± 0.080 [b] | <0.0001 | 0.1 ± 0.02 | 0.0 ± 0.01 |
| 22:4n6 | Adrenic acid | 1.7 ± 0.17 [a] | 0.6 ± 0.16 [b] | 2.6 ± 0.33 [c] | <0.0001 | 2.1 ± 0.22 | 1.5 ± 0.07 |
| 22:5n6 | n6-Docosapentaenoic acid | 0.72 ± 0.119 | 0.48 ± 0.114 | 1.10 ± 0.228 | 0.0566 | 0.8 ± 0.29 | 0.3 ± 0.02 |
| 22:5n3 | n3-Docosapentaenoic acid | 1.16 ± 0.137 | 0.84 ± 0.132 | 1.50 ± 0.263 | 0.0688 | 1.9 ± 0.20 | 1.1 ± 0.07 |
| 22:6n3 | Docosahexaenoic acid | 2.0 ± 0.26 | 2.3 ± 0.24 | 2.0 ± 0.49 | 0.6429 | 1.7 ± 0.42 | 1.7 ± 0.11 |
| 24:0 | Lignoceric acid | 0.08 ± 0.017 [a] | 0.05 ± 0.017 [a] | 0.17 ± 0.033 [b] | 0.0217 | 0.0 ± 0.01 | 0.1 ± 0.01 |
| 24:1 | Nervonic acid | 0.12 ± 0.042 [a] | 0.32 ± 0.040 [b] | 0.21 ± 0.081 [a,b] | 0.0108 | 0.26 ± 0.03 | 0.35 ± 0.02 |

| | Cold-Stunned Sea Turtle Fatty Acid Profiles | | | | Published Healthy Sea Turtle Fatty Acid Profiles | |
|---|---|---|---|---|---|---|
| *Fatty Acid Groups* | *Species* | | | *p*-Value [#] | *Species* | |
| | *C. mydas* (n = 11) | *L. kempii* (n = 12) | *C. caretta* (n = 3) | | *C. mydas* (n = 11) | *L. kempii* (n = 3) |
| Highly Unsaturated Fatty Acids (HUFA) | 21.8 ± 1.18 [a] | 18.1 ± 1.13 [b] | 24.0 ± 2.26 [a] | 0.0325 | 22.7 ± 1.65 ^ | 20.8 ± 0.67 ^ |
| Monoenes | 33.7 ± 0.99 [a] | 41.6 ± 0.95 [b] | 40.1 ± 1.89 [b] | <0.0001 | 37.5 ± 1.67 ^ | 43.0 ± 0.76 ^ |
| Total n3 fatty acids | 6.4 ± 0.59 | 8.3 ± 0.57 | 7.1 ± 1.14 | 0.0746 | 7.4 ± 0.67 | 7.6 ± 0.53 |
| Total n6 fatty acids | 21.5 ± 1.15 [a] | 14.7 ± 1.10 [b] | 21.7 ± 2.20 [a] | 0.0006 | 21.4 ± 1.75 | 17.6 ± 0.63 |
| Total n9 fatty acids | 25.6 ± 0.79 [a] | 21.9 ± 0.75 [b] | 25.5 ± 1.51 [a] | 0.0054 | 29.3 ± 1.41 | 22.5 ± 0.13 |
| n-6:n-3 Fatty acid ratio | 3.9 ± 0.49 [a] | 1.9 ± 0.47 [b] | 3.0 ± 0.93 [a,b] | 0.0186 | 3.1 ± 0.37 ^ | 2.4 ± 0.17 ^ |
| Polyunsaturated fatty acids (PUFA) | 27.9 ± 1.26 [a] | 23.0 ± 1.21 [b] | 29.0 ± 2.42 [a] | 0.0163 | 29.3 ± 1.46 ^ | 25.2 ± 0.74 ^ |
| Saturates | 38.4 ± 0.77 [a] | 35.4 ± 0.74 [b] | 30.8 ± 1.47 [c] | 0.0003 | 33.2 ± 0.64 | 31.5 ± 1.78 |

* Fatty acids lauric acid (12:0), pentadecenoic acid (15:1), myristoleic acid (16:1n5), and 13-octadecenoic acid (18:1n5) were analyzed but did not have detectable values. [#] P-values in bold are significant and each fatty acid without similar superscripts (a,b,c) per row differs. ^ Indicated fatty acid grouping totals were not reported in prior publication[22] and were calculated from the original data to enable comparison to the present study.

Significant differences among species were identified for 19 out of 36 individual fatty acids analyzed and six out of seven fatty acid groups evaluated ($p < 0.5$). The n3 fatty acids

were the only group for which there was no significant difference among species. Of the remaining groups, highly unsaturated fatty acids (HUFA), n6 fatty acids, n9 fatty acids, and polyunsaturated fatty acids (PUFA) were significantly higher for cold-stunned green and loggerhead in comparison to cold-stunned Kemp's ridley sea turtles. Monoenes alone were significantly higher for cold-stunned Kemp's ridley and loggerhead sea turtles compared to cold-stunned green sea turtles. Significant differences in saturated fatty acid proportions were found among all three species with the highest proportion found in green sea turtles, followed by Kemp's ridley and loggerhead sea turtles, respectively. The n6:n3 fatty acid ratio differed significantly between cold-stunned green and Kemp's ridley sea turtles, at 3.9 and 1.9, respectively; no significant differences in the n6:n3 ratio were identified between cold-stunned loggerhead sea turtles and either of the other species.

Across all three species, palmitic acid (16:0) and stearic acid (18:0) were the major saturated fatty acids, with palmitic acid predominating; oleic acid (18:1n9) was the predominant monoene; and arachidonic acid (20:4n6) was the predominant PUFA. Palmitoleic acid (16:1n7) was also a major contributor to the total proportion of monoenes in cold-stunned kemp's ridley and loggerhead sea turtles, which had, respectively, more than triple and double the relative amount of this fatty acid than cold-stunned green sea turtles. Arachidonic acid and oleic acid proportions were significantly higher in cold-stunned green sea turtles in comparison to cold-stunned Kemp's ridley sea turtles; the proportions of these two fatty acids were not significantly different between loggerhead sea turtles and either of the other species. No significant differences among species were found for essential fatty acids linoleic acid (18:2n6) and $\alpha$-linolenic acid (18:3n3). The proportions of long-chain marine n3 PUFA eicosapentaenoic acid (EPA, 20:5n3) in cold-stunned Kemp's ridley sea turtles were significantly higher compared to cold-stunned green sea turtles; no significant differences in EPA were identified between cold-stunned loggerhead sea turtles and either of the other species. No significant differences among species were identified for long-chain marine n3 PUFA docosahexaenoic acid (DHA, 22:6n3).

Upon visual comparison, the whole blood fatty acid profiles of cold-stunned green and Kemp's ridley sea turtles were similar to published profiles of healthy conspecifics in terms of both differences between species and the quantitative proportions of individual fatty acids and fatty acid groups [22]. All fatty acids accounting for at least 2% of the total fatty acids present differed by less than 21% between cold-stunned and healthy green sea turtles and less than 29% between cold-stunned and healthy Kemp's ridley sea turtles.

Despite the similarities, quantitatively minor but apparent differences were observed between cold-stunned and healthy animals. Cold-stunned green and Kemp's ridley sea turtles had marginally lower total proportions of PUFA and monoenes and higher total proportions of saturated fatty acids than healthy conspecifics [22]. Highly unsaturated fatty acids (HUFA, a subset of PUFA) were also lower in cold-stunned relative to healthy green sea turtles, but not in cold-stunned relative to healthy Kemp's ridley sea turtles. The mean proportions of the major parent fatty acids of the n6 and n3 groups, linoleic acid (18:2n6) and $\alpha$-linolenic acid (18:3n3), were slightly lower in cold-stunned relative to healthy green sea turtles; a similar difference was not noted for Kemp's ridley sea turtles. The mean proportion of arachidonic acid (20:4n6) was slightly higher in cold-stunned relative to healthy green sea turtles and slightly lower in cold-stunned relative to healthy Kemp's ridley sea turtles. For both species, the mean proportion of oleic acid (18:1n9) was lower in cold-stunned relative to healthy sea turtles. The proportions of long-chain marine n3 PUFA DHA (22:6n3) and EPA (20:5n3) in both cold-stunned green and Kemp's ridley sea turtles were slightly higher and similar, respectively, to the proportions detected in healthy conspecifics.

## 5. Discussion

The proportions of more than half of the individual fatty acids analyzed and all but one fatty acid group evaluated differed significantly among sea turtle species in the present study examining cold-stunned juveniles. Given the pronounced differences in

the feeding strategies of these species [37], this finding is not surprising, and similar species differences were observed between presumed healthy juvenile green and Kemp's ridley sea turtles in both wild and managed-care settings [22,23]. Both Kemp's ridley and loggerhead sea turtles are primarily carnivorous throughout their lifetime, though they do exhibit differences in prey selection [37]. In contrast, while predominantly carnivorous as hatchlings, juvenile green sea turtles transition to a primarily herbivorous diet that they maintain into adulthood [37,38]. Only three loggerhead sea turtles were included in the analysis; thus, limited conclusions can be drawn from the data for this species. As whole blood fatty acid profiles of loggerhead sea turtles have not previously been published to the authors' knowledge, these data provide novel baseline information.

Findings that total n6 fatty acids were considerably higher in cold-stunned green sea turtles compared to cold-stunned Kemp's ridley sea turtles and total n3 fatty acids were similar among species were consistent with data from healthy free-ranging conspecifics [22]. The major parent compounds of the n6 and n3 fatty acids groups are linoleic acid (18:2n6) and $\alpha$ -linolenic acid (18:3n3), which are both considered essential fatty acids as they cannot be synthesized de novo in most vertebrates and must be obtained through dietary intake [39,40]. Linoleic acid is the metabolic precursor of arachidonic acid (20:4n6), a key modulator of inflammatory responses and a substrate for the cyclooxygenase (COX) enzyme system. Arachidonic acid was present in relatively large quantities in all sea turtle species. In terrestrial mammalian obligate carnivores, arachidonic acid is considered essential because they cannot elongate and desaturate linoleic acid to arachidonic acid [41]; it is unknown if this is the case for carnivorous sea turtles. Arachidonic acid can also become conditionally essential in other vertebrates if deficiencies in linoleic acid, or the enzymes needed to convert it to arachidonic acid, exist [39]. While cold-stunned green sea turtles had higher percentages of linoleic acid, $\alpha$-linolenic acid, and arachidonic acid than cold-stunned Kemp's ridley sea turtles, only the difference for the latter was statistically significant. This contrasts with healthy free-ranging animals, in which green sea turtles had significantly higher percentages of linoleic acid and $\alpha$-linolenic acid than Kemp's ridley sea turtles, but no significant difference in arachidonic acid proportions was identified between species [22]. Despite this, percentages of these three key fatty acids were clinically similar between cold-stunned sea turtles and healthy free-ranging conspecifics.

While arachidonic acid (20:4n6) was present in relatively high quantities in both cold-stunned and healthy free-ranging sea turtles, markedly lower quantities were reported in green and Kemp's ridley sea turtles undergoing rehabilitation, at approximately 10% and 20% those of wild conspecifics, respectively [22,23]. The detection of consistently high levels of this conditionally essential fatty acid in two cohorts of wild sea turtles sampled in different years and seasons highlights that the modification of sea turtle diets in managed-care settings may be warranted to ensure the adequate intake of arachidonic acid [22,23]. Of further interest, the percentages of linoleic acid (18:2n6) and $\alpha$-linolenic acid (18:3n3) were lower in green and Kemp's ridley sea turtles undergoing rehabilitation relative to both cohorts of wild conspecifics; thus, these essential fatty acids may also be a point of concern for sea turtle diet design in managed care [22,23].

In cold environments, poikilothermic species must increase the proportion of unsaturated fatty acids (monoenes and PUFA) in cell membranes to maintain membrane fluidity [42–45]. The long-chain n3 PUFA docosahexaenoic acid (DHA, 22:6n3) plays a preeminent role in this process, and as such, cold-water marine species often require a higher dietary intake of this fatty acid [46,47]. Another influential marine long-chain n3 PUFA is eicosapentaenoic acid (EPA, 20:5n3). EPA, and DHA to a lesser extent, are important anti-inflammatory immunomodulators [48]. It is interesting that all sea turtle species had low levels of DHA and EPA and high levels of arachidonic acid (20:4n6) relative to other cold-water marine animals, such as fish and cetaceans [24,47,49,50]. As EPA and DHA compete directly with arachidonic acid to limit the production of pro-inflammatory eicosanoids [48], the relative proportions of these fatty acids may be important to consider in future research investigating the pathogenesis of cold-stunning. In contrast to both co-

horts of free-ranging conspecifics (healthy and cold-stunned), green and Kemp's ridley sea turtles in rehabilitation had notably higher levels of both EPA and DHA than arachidonic acid. More research is warranted to clarify the role of these long-chain PUFA in sea turtle nutrition and the potential implications for diet design in managed care.

Given that cold-stunned sea turtles are anorexic and subject to intense physiologic and metabolic challenges, it seems probable that they would have altered fatty acid status; however, the discrepancies noted between cold-stunned and healthy cohorts were marginal. One potential explanation for this is the limited contribution of plasma fatty acids to the total fatty acids present in whole blood. As previously mentioned, whole blood fatty acid profiles are derived from a blend of erythrocyte and plasma lipids, which are indicative of long-term and short-term fatty acid status, respectively, in mammals [29,31]. Lipid turnover in reptile erythrocytes, which are nucleated and have exceptionally long lifespans (~600–800 days compared to ~120 days in humans) [51], could differ considerably from mammalian erythrocytes, but to the authors' knowledge, this has not been thoroughly investigated. While the exact durations of cold-stunning episodes for the sampled sea turtles are unknown, based on weather patterns and physical examinations, the investigators strongly suspect that these were acute events (generally < 7 days). Additionally, postmortem examinations of non-surviving sea turtles in the same cold-stun cohorts typically had ample intestinal contents and sometimes stomach contents, indicating feeding up until the acute event (C. Harms, pers. obs.). Thus, it is unlikely that changes in fatty acid status due to cold-stunning would have any measurable impact on erythrocyte fatty acid composition. The general congruency seen between the whole blood fatty acid profiles of cold-stunned sea turtles and healthy conspecifics supports the expectation that an appreciable proportion of fatty acids in dried whole blood spot samples are derived from erythrocytes and reflect long-term status.

Acute changes in circulating plasma fatty acids alone during cold-stunning episodes may account for the marginally lower proportions of total PUFA, total monoenes, and several key unsaturated fatty acids detected in cold-stunned relative to healthy sea turtles. This may reflect in vivo depletion of these compounds during hypothermia, or, given that cold-stunned sea turtles are anorexic, ceased dietary intake of unsaturated fatty acids may synergistically or alternatively account for these discrepancies. However, literature, including several studies in reptiles, suggests that decreases in total saturated fatty acids and increases in total unsaturated fatty acids would be expected with fasting, which is the opposite of what was seen in cold-stunned relative to healthy sea turtles [52–54].

Alternatively, the differences observed between cold-stunned and healthy sea turtles could simply be the result of natural temporal variation, as the abundance of prey items and the dietary intake of sea turtles can vary seasonally and annually [37]. Relatedly, sea turtles might also experience seasonal changes in fatty acid metabolism as an adaptive response to changing temperatures. Sea turtle diets can also vary with age [37], and while the exact ages of the sea turtles included in this study are unknown, all sea turtles were of a juvenile life stage and were similar in size and presumed age to previously sampled healthy individuals.

Though a seemingly less-probable explanation, the marginally lower proportions of PUFA in cold-stunned sea turtles could potentially represent an adaptive mechanism against the development of steatitis. A small percentage of cold-stunned Kemp's ridley sea turtles have been diagnosed with steatitis weeks to months into rehabilitation [55]. Steatitis can occur during starvation and has also been reported in a variety of species fed diets high in PUFA and deficient in vitamin E, a potent antioxidant [56–62]. The putative explanation for this condition is that when free radicals are produced from the enhanced metabolism of PUFA in the absence of sufficient vitamin E, fat is oxidized.

The primary limitation of interpreting these results is the uncertainty surrounding the relative contributions of plasma-based and cellular lipids to total fatty acid content in whole blood, or rather the degree to which the whole blood fatty acid profiles of sea turtles can reflect acute changes in fatty acid status. Fatty acid analysis of plasma samples from

cold-stunned sea turtles could allow for better understanding of the immediate impacts of cold-stunning on circulating fatty acids and might reveal more pronounced differences between cold-stunned and healthy sea turtles; however, the required sample processing is less convenient for use in field settings than dried blood spot cards.

## 6. Conclusions

Whole blood fatty acid profiles of cold-stunned green and Kemp's ridley sea turtles largely resembled those of presumed healthy conspecifics. Marginal differences included lower proportions of PUFA and monoenes and higher proportions of saturates in cold-stunned relative to healthy sea turtles. The lower proportions of unsaturated fatty acids in cold-stunned sea turtles may reflect acute changes in the plasma-based component of the whole blood fatty acid signal. Alternatively, natural seasonal variations in diet and/or metabolism could explain the observed differences. The congruency between fatty acid profiles of cold-stunned and healthy free-ranging sea turtles supports the expectation that a considerable proportion of the fatty acid signal in whole blood reflects long-term fatty acid status. Comparisons with plasma and erythrocyte fatty acid profiles are needed to ascertain whether acute changes in circulating fatty acid status in sea turtles can be elucidated using whole blood samples.

**Author Contributions:** Contributions are as follows. Conceptualization, L.J.M., C.A.H. and K.D.A.-v.H.; methodology, L.J.M., C.A.H. and K.D.A.-v.H.; software, D.B. and K.D.A.-v.H.; validation, D.B. and K.D.A.-v.H.; formal analysis, A.C.H., L.J.M., C.A.H., E.A.K. and K.D.A.-v.H.; investigation, A.C.H., L.J.M., C.A.H., E.A.K. and K.D.A.-v.H.; resources, L.J.M., C.A.H., D.B. and K.D.A.-v.H.; data curation, A.C.H., L.J.M., C.A.H. and K.D.A.-v.H.; writing, A.C.H., L.J.M., C.A.H. and K.D.A.-v.H.; visualization, L.J.M., C.A.H. and K.D.A.-v.H.; supervision, C.A.H. and K.D.A.-v.H.; funding acquisition, L.J.M., C.A.H. and K.D.A.-v.H.; All authors have read and agreed to the published version of the manuscript.

**Funding:** This research was funded by the North Carolina Zoo.

**Institutional Review Board Statement:** Animal capture, sample collection, and transfer were authorized by the North Carolina Wildlife Resources Permit 21ST42 and 22ST42 and North Carolina State University IACUC protocol 20-166-01.

**Data Availability Statement:** The data presented in this study are available via the corresponding author upon reasonable request.

**Acknowledgments:** The authors thank all individuals responsible for the rescue, rehabilitation, and release of the sea turtles enrolled in this study.

**Conflicts of Interest:** The authors declare no conflict of interest.

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
