# Peer review of "Whole Blood Fatty Acid Profiles of Cold-Stunned Juvenile Green, Kemp’s Ridley, and Loggerhead Sea Turtles"

_2673-5636, doi:10.3390/jzbg4010001_

Round 1

Reviewer 1 Report

The main contribution is to guide readers on the main conditions of tropical marine species that move away from their natural space due to environmental variation. Metabolic tests are a fundamental tool for the understanding of sudden clinical pictures, contributing to the recognition of reference values for the interpretation of the metabolic profile of fatty acids of species with a high energy expenditure undoubtedly contributes to have better opportunities in the timely care of these conditions 

Subtropical species have a higher concentration of essential polyunsaturated fatty acids than tropical species, although biotic and abiotic factors significantly influence their concentration. The values presented may change according to modifications in the ecosystem and in the characteristics of each species, however, the knowledge generated in this work is useful to understand these risks. 

Reviewer 2 Report

Based on the results of this study, it would be interesting to examine the antioxidant status of cold-stunned sea turtles in some future studies.
